



# On the Response of the Equatorial Atmosphere and Ocean to Changes in Sea Surface Temperature along the Path of the North Equatorial Counter Current

David John Webb[1]

[1]National Oceanography Centre, Southampton SO14 3ZH, U.K.

**Correspondence:** David John Webb (djw@noc.ac.uk)

**Abstract.** The CESM climate model is used to test the hypothesis that changes observed during El Niños are, at least in part, a response of the coupled ocean/atmosphere system to changes in sea surface temperature along the path of the North Equatorial Counter Current.

The results from the second month in a set of forced runs show that increased temperatures at the latitudes of the North

Equatorial Counter Current produce a significant increase in deep atmospheric convection within the Intertropical Convergence Zone. This has a local effect on the ocean's surface pressure field which reduces pressures on the Equator. The increased deep atmospheric convection also affects the longitude structure of the Hadley Circulation. In the south-east Pacific, an area associated with Hadley Cell sinking, surface pressure decreases. In the western Pacific, the pressure field increases with maxima north and south of the Equator.

Together the surface pressure changes have similarities with those associated with the Southern Oscillation. They reduce the zonal component of wind stress along the Equator and produce an El Niño type response in the ocean.

## 1 Introduction

During a period when studies of El Niños generally concentrated on the seas close to South America, Wyrtki (1973, 1974) used sea level data from Pacific Islands to show that, in the western Pacific, the transport of the North Equatorial Counter Current

(NECC) increased during El Niño events. His studies led to a wider interest in the subject and eventually to our understanding that El Niños are responsible for events both locally in the Pacific and at global scales (McPhaden et al., 2020).

Since those early studies there has been a huge amount of research on the El Niño and related problems, much of which is also summarised in McPhaden et al. (2020). In the atmosphere, many of the studies focus on the generation and propagation of Rossby waves, which interact with the jet streams and affect weather patterns over large parts of the globe (Trenberth et al.,

1997).

The Rossby waves are often described as being generated by the high-level divergences produced by deep atmospheric convection near the Equator (Sardeshmukh and Hoskins, 1988). However, theoretically it is difficult to generate such waves in the equatorial easterly winds usually found near the Equator, a result supported observationally by Rasmussen and Mo (1993)





who found only weak Rossby wave sources within 15 degrees of the Equator. Instead they found that the strongest Rossby wave sources were in the regions of convergence generated by the descending branch of the Hadley Cell.

In the ocean, the focus has been on the equatorial band and changes in the structure of the thermocline and equatorial currents. In a normal year, easterly winds over the Equator generate an east to west slope in sea level and a similar increase in depth of both the surface thermocline and the depth of the Equatorial Undercurrent. The winds also cause upwelling along the Equator, generating the eastern Pacific cold pool.

During El Niño events, the easterly wind weakens, there is a relaxation of thermocline slope, a change in the undercurrent, and reduced upwelling on the Equator resulting in a warmer cold pool.

West of the dateline, the easterlies near the Equator are replaced by westerlies. The resulting Ekman transport then converges warm pool water onto the Equator where the wind now generates an eastward flowing Equatorial Current. The current carries the warm water towards the dateline where a region of deep atmospheric convection develops.

A number of mechanisms have been proposed as the cause of El Niños. A common assumption is that the El Niño-La Niña cycle is an unstable mode of the coupled ocean-atmosphere system. In the ocean this is often described in terms of a discharge-recharge mode (Wyrtki, 1975, 1985; Jin, 1997; Chakravorty et al., 2021), the warm pool-of the west Pacific being discharged during each El Niño event and then being recharged in the years preceding the next event.

Early theories proposed that the events were triggered by equatorial Kelvin and Rossby waves in the ocean, but at present the two main contenders are Madden-Julian oscillations (Slingo et al., 1999; Saith and Slingo, 1999; Li et al., 2023) and westerly wind bursts (McPhaden et al., 1981; Delcroix et al., 1993; Yu and Fedorov, 2020).

Both types of event have been observed in the western Pacific prior to El Niños and both could move warm pool water towards the dateline. But as stated by Yu and Ferorov (2022), when discussing westerly wind bursts, "quantifying their effects is challenging".

Published research on El Niños usually concentrates on the equatorial band and there have been few papers on the NECC and the nearby Intertropical Convergence Zone (ITCZ). However, as well as Wyrtki's early papers, Rasmussen and Carpenter (1982) reported changes in the winds associated with the ITCZ and its southward movement during an El Niño.

Others also studied changes to the NECC and the ITCZ during El Niño events but, although correlations can only provide a statistical connection, not a causal one, the discussion is primarily in terms of the passive response of the NECC and the ITCZ to the El Niño signal (i.e. Zhao et al. (2013), Zhou et al. (2021)), not their possible active role.

## 1.1 The Present Paper

The present paper is a development from Webb (2018b), a study of the equatorial currents during the strong El Niños of 1982-83, 1997-98 and 2015-16. The study used data from a high-resolution ocean model and found that the NECC was much warmer than normal during the development of strong El Niños. The study also showed that the warmer than normal water came from the warm pool in the western Pacific and that it was carried eastwards by a stronger than normal NECC, starting in the west during the northern spring and arriving in the eastern Pacific late in the year.





Although the discharge part of the discharge-recharge mechanism is often described in terms of gravity driven currents along the Equator, this study indicated that in the cases studied, much of the west Pacific warm pool was advected out of the region by the geostrophically balanced North Equatorial Counter Current (Webb, 2018b, Fig. 13).

A follow up study using satellite data (Webb et al., 2020) confirmed the findings, and a second modelling study (Webb, 2021) showed that it was the winds in the western Pacific early in the year which were responsible for the initial increase of transport by the NECC.

This connection between the winds in the western Pacific early in the year and the development of a strong El Niño late in the year could be explained if it is the warm NECC itself which is at least partly responsible for the strong El Niño. The time

required to advect water across the Pacific would also help explain the long timescales involved in the fluctuations of El Niño indices.

## 1.2  Hypothesis

Based on the comments above, it is tempting to hypothesise that strong El Niños are due, at least in part, to a warmer NECC. However, extreme events may be unstable in other ways, so here the focus is on a less extreme state of the ocean, testing the

hypothesis that "changes observed during El Niños are, at least in part, a response of the climate system to changes in sea surface temperature along the path of the North Equatorial Counter Current".

The study also tests a hypothesis that the close relationship between the NECC and ITCZ is involved. The NECC lies just south of the latitude of the ITCZ. Convection in the ITCZ draws in surface air from the north and south, but that from the south crosses the NECC and so is readily affected by any changes in sea surface temperature.

Sea surface temperatures (SST) on the Equator in the central and eastern Pacific are usually too cold trigger deep atmospheric convection. However, those in the NECC are usually above 26° C, close to the temperature range in which deep atmospheric convection is observed over the ocean (Gadgil et al., 1984; Evans and Webster, 2014; Kubar and Jiang, 2019). In addition, the remoteness from other centres of deep convection may make the region more sensitive than normal to changes in SST (Williams et al., 2023).

The energy available to drive deep convection also depends critically on atmospheric water content. Over the ocean relative humidity is high, around 80%, so the total water content is, to first order, a function of temperature, a one-degree rise increasing the water content by 7% or more (Biri et al., 2023). As a result, even small increases in NECC temperature can have a significant effect on convection in the nearby ITCZ.

The effect of convective cells at and close to the Equator was first discussed by Matsuno (1976) and later Gill (1980), using

an analytic model which represented the first vertical mode of the atmosphere. Unfortunately their model only considers the case of convective cells with small horizontal scale, whereas the ITCZ extends for thousands of kilometres.

A convective source, closer in structure to that of the ITCZ, is discussed by Vallis (2017). In Fig. 8.16 he shows the first mode solution for an infinitely long line of convection at a fixed latitude. With no background wind field, the solution shows that there is a band of westerly winds to the south of the line of convection. The width of the band depends on the Equatorial



Rossby radius $L_{eq}$, and the time scale for development depends on $T_{eq}$ where,

$$L_{eq} \quad = \quad (c/(2\beta))^{1/2}, \tag{1}$$
$$T_{eq} \quad = \quad (2c\beta)^{1/2}. \tag{2}$$

Here $c$ is the speed of the first (and fastest) vertical mode of the atmosphere and $\beta$ is the gradient of the Coriolis term with latitude. If $c$ is $25\ \mathrm{m\,s^{-1}}$ then $L_{eq}$ corresponds to around 7 degrees of latitude and $T_{eq}$ corresponds to about 8 hours. If the 95 estimates are correct, then increases in ITCZ convection a few degrees north of the Equator will affect winds on the Equator within a few hours.

Given the strong easterly winds normally found on the Equator, the changed pressure field may not be sufficient to change the direction of the winds, but even a small reduction in wind stress would affect the east-west surface slope of the ocean, and so may produce other El Niño type changes.

As a thought experiment, with simple quasi-linear logic, this seems reasonable. But the climate system can be highly non-linear, so the hypothesis needs testing in a more realistic environment. In the present study this is achieved by setting up a series of response experiments using the NCAR Community Earth System Model (CESM). A simple forcing is used in which the temperature along part of the NECC is increased slightly.

The first of the experiments using the model is then used for a detailed analysis of the processes involved. This involves fol-105 lowing the resulting changes in ITCZ convection and the changes in atmospheric pressure, equatorial winds and the equatorial ocean, following the chain of events proposed above. The variability due to changes in the initial condition in the equatorial Pacific and elsewhere is then studied using a series of ten additional experimental runs.

The overall study does not check that the NECC is responsible for the timescales of El Niños, but it does check that a typical NECC temperature fluctuation is sufficient to produce a recognisable El Niño signal.

In the rest of the paper, section 2 gives details of the CESM model and the relaxation term used in the forced run. Sections 3 and 4 then report on the response of the atmosphere and ocean in the first of the experiments.

Section 5 summarises the variability of key indices in the full set of experimental runs and makes comparisons with results from the ECMWF ERA5 atmospheric reanalysis project (Hersbach et al., 2022). Finally, section 6 discusses the wider implications of the study.

## 2  Model and Forcing Experiment

This study makes use of version 2.1.3 of CESM, the NCAR Community Earth System Model (Hurrell et al., 2013; Lauritzen et al., 2018). The model is widely used by the earth science community for global change research (Schneider et al., 2022; Holland et al., 2024), and for studies of individual parts of the climate system (Li et al., 2018; Dolores-Tesillos et al., 2022; Maher et al., 2023).





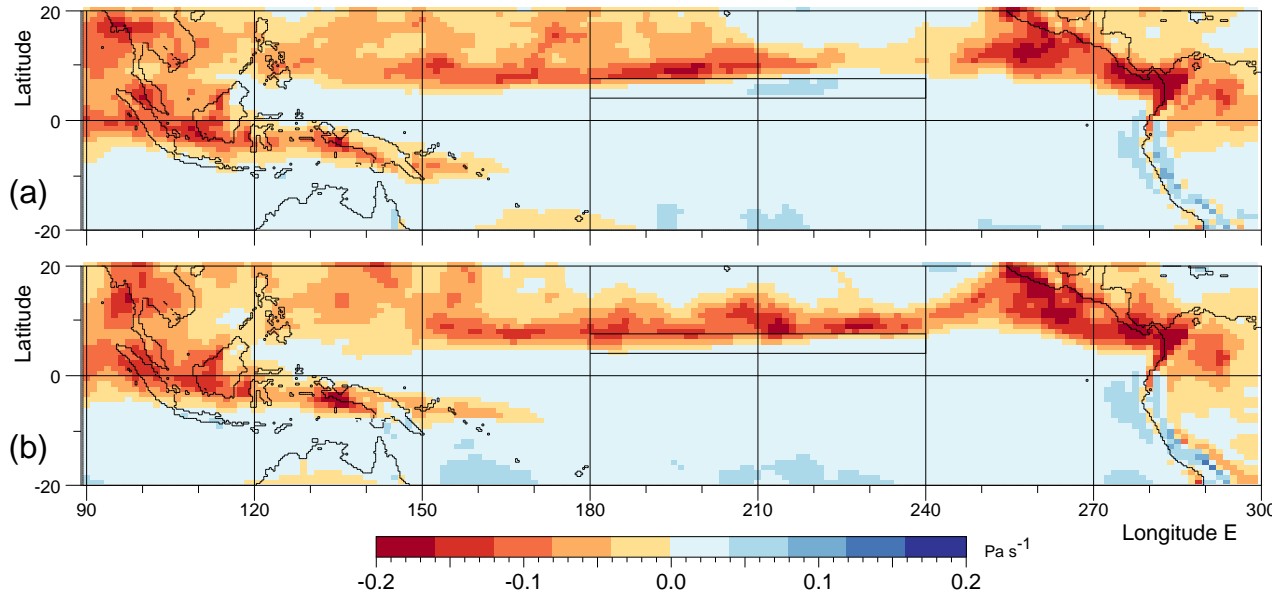

**Figure 1.** Average rate of change of pressure following particles ($dP/dt$), also known as Omega, crossing the 525 hPa pressure surface in September of year 5 for (a) the control run and (b) the forced run. The forcing, described in the text, starts at the beginning of August in year 5 and is non-zero within the marked rectangular region in the central Pacific. Negative (red) values correspond to convection.

The model code is supported by a large number of component sets, each designed for use in a different type of climate study. The component set used for the present study is 'B1850', a basic set in which the sub-models representing atmosphere, ocean, ice and land are all active. The final sub-model, representing ocean surface waves, uses a fixed wave field.

For the present study, the two key components are the ocean and the atmosphere. The atmosphere has 32 levels in the vertical, and resolutions of 1.25 degrees in longitude and 0.934 degrees in latitude. The ocean has 60 layers in the vertical and a longitude resolution of 1.1 degrees. Within 20 degrees of the Equator the latitude resolution is 0.26 degrees, extending to 1.125 degrees nearer the poles. The model layers have a thickness of 10 m in the top 200 m of ocean after which the thickness increases with depth.

The ocean model resolution means that it should be able to represent most of the small-scale features of the tropical ocean, but the longitude resolution may be too large for a good representation of tropical instability waves (Graham, 2014). The model produces a reasonable El Niño response (Capotondi et al., 2020) but, like many other similar models, its Niño 3.4 index does not show the asymmetry between El Niño and La Niña states, seen in reality.

Many coupled climate models suffer from the double ITCZ problem (Mechoso et al., 1995; Song and Zhang, 2019; Tian and Dong, 2020), making comparison against observations difficult. The present short runs of CESM model, version 2.1.3, showed no evidence of a double ITCZ.




## 2.1 Forcing Experiment

The present study is based on a control run and a forced run. In the control run, the model was started with the default B1850 initial fields and parameters, and run for a period of 10 years. At the end of each month, the model archived the previous month's averages of key variables. It also generated a set of restart files for the following month.

Although the original hypothesis was developed after studying strong El Niños, it was thought better to start with a test during an average year when other components of the climate were less likely to trigger an El Niño type response. During the first eighteen months of the control run the Niño 3.4 index was high, around 2 degrees, but by year 5 it had dropped to values near zero. Year 5 was therefore chosen for the test.

Previous studies (Webb, 2018b; Webb et al., 2020; Webb, 2021) showed that during the development of strong El Niños, the period around September appeared to be an important growth period, with NECC temperatures around 30° C between 180° E and 240° E (120° W). The forced run was therefor initialised using the control run restart data for the 1st of August in year 5 and continued for a period of five months. The forcing was limited to the rectangular region, shown in Fig. 1, lying between 180° E and 240° E and between 4° N and 7.5° N. In the control run, there was an east-west slope in SST, the average temperatures at 180° E and 240° E being around 28.5° C and 27.5° C respectively.

The forced run was designed to increase the sea surface temperature in the region by approximately 1° C, a value typical of the observed changes in the region (Fig. B1). The increase was achieved using a forcing function independent of latitude but a linear function of longitude, with values of 29.5° C and 28.5° C at its western and eastern limits.

The forcing was applied to the top 100 m of ocean. In the top 30 m, the forced run used a decay time of two days. Below this, the relaxation coefficient, the inverse of the decay time, decreased linearly with depth to a value of zero at 100 m.

The forcing region is slightly to the south of the path of the NECC in the control run but corresponds to the latitude of the NECC during a strong El Niño (Webb, 2018b). During August the average surface temperature within the forcing region is 28.81° C in the forced run, and 27.75° C in the control run, while in September the values are 28.91° C and 27.54° C.

## 3 Response of the Atmosphere

The results, presented in the next two sections, focus on the average response of the atmosphere and ocean during September of year five. The following months showed a steady increase in the CESM monthly Niño 3.4 index, instead of the fall shown by the control run, but September is chosen because, by the centre of the month, the main atmospheric response should have settled down and there will have been sufficient time for the ocean response to develop. Seasonal changes should also not be a major factor in the response.

## 3.1 Atmospheric Convection

Figure 1 shows the the rate of change of atmospheric pressure following particles crossing the 525 hPa surface in the control and forced runs. Negative values correspond to regions of convection and positive values to regions of sinking.





The ITCZ shows as a band of enhanced convection north of the Equator, bounded roughly by longitudes 150° E and 250° E (110° W). In the east it merges with the area of convection lying above the East Pacific Warm Pool.

The result from the control run is in reasonable agreement with results from the ECMWF ERA5 reanalysis (Webb, 2025)[1]. Convection over the ocean to the east of New Guinea is weaker than expected and the gap in the ITCZ around 240° E (120° W) may also be unusual.

The area in the central Pacific, where the temperature is modified during the forced run, is shown by the rectangle lying north of the Equator. The results show that in the forced run, middle-atmosphere convection in the nearby ITCZ moves south and increases, especially between 200° E (160° W) and 240° E (120° W), closing the gap between the ITCZ to the East Pacific Warm Pool. North of 12° N, convection is reduced between 160° E and 210° E (150° W) but increases further west.

### 3.1.1 Quantitative Fluxes

If the atmosphere is in hydrostatic equilibrium, then the vertical pressure gradient $dP/dz$ is given by

$$dP/dz \quad = \quad -g\,\rho, \tag{3}$$

where $g$ is the standard gravity, $\rho$ is the density of air and $P$ is pressure. If the vertical velocity relative to the surfaces of constant pressure (which themself are normally moving) is $w_L$, then,

$$w_L \quad = \quad \mathrm{d}P_L/\mathrm{d}t \,/\, \mathrm{d}P/\mathrm{d}z,$$
$$= \quad -(1/(g\rho))\,\mathrm{d}P_L/\mathrm{d}t, \tag{4}$$

where $P_L$ is the pressure experienced by a Lagrangian particle moving with the flow, $t$ is time and $z$ is height. If $F_L$ is the vertical mass flux across the moving surfaces of constant pressure, then,

$$F_L \quad = \quad \rho\,w_L,$$
$$= \quad -(1/g)\,\mathrm{d}P_L/\mathrm{d}t. \tag{5}$$

Integrating over an area, the total flux $F$, is given by,

$$F \quad = \quad -(1/g)\ \int (dP_L/dt)\,R^2\,cos(\phi)\,d\psi\,d\phi, \tag{6}$$

where $\psi$ is longitude, $\phi$ latitude and $R$ the radius of the Earth.

A second integral, $F_{W>0}$, includes only the convecting regions,

$$F_{W>0} \quad = \quad -(1/g)\ \int (dP_L/dt)\,\theta(-dP_L/dt)$$
$$R^2\,cos(\phi)\,d\psi\,d\phi, \tag{7}$$

where,

$$\theta(x) \quad = \quad 1 \quad if \ \ x > 0,$$
$$= \quad 0 \quad otherwise$$



**Table 1.** Average integrated vertical mass fluxes ($\mathrm{Tg\,s^{-1}}$) in the control and forced runs during September of year 5. The integrals are between the Equator and 20° N. Different columns correspond to either the forcing longitudes (180° E to 240° E (120° W)) or the whole latitude band (0° E to 360° E) and are for either the full integral or for the integral only over convecting regions where $dP_L/dt$ is less than zero, $P_L$ being the pressure following a particle.

| Level | Run | 180° E-240° E | | 0° E-360° E | |
|---|---|---|---|---|---|
| hPa | | Total | $dP_L/dt < 0$ | Total | $dP_L/dt < 0$ |
| 322 | Control | 11.9 | 32.8 | 248.5 | 330.5 |
| | Forced | 26.6 | 48.9 | 240.4 | 320.4 |
| 525 | Control | 16.5 | 38.1 | 260.1 | 334.2 |
| | Forced | 32.1 | 48.9 | 255.8 | 327.9 |
| 821 | Control | 51.4 | 73.3 | 254.7 | 330.4 |
| | Forced | 50.0 | 69.7 | 250.1 | 320.3 |

A summary of the fluxes in the two runs and their variation with height is given in Table 1. Here the latitude integral is between the Equator and 20° N, the range chosen to include the whole of the rising branch of the Hadley Cell during the late northern summer. The integral over longitude is either for the full 360 degrees or for just the forcing region between 180° E and 240° E (120° W).

    In the control run, the fluxes in the forcing region almost halve between 821 hPa and the middle atmosphere at 525 hPa.

There is then a small drop to 322 hPa, just below the tropopause. This contrasts with the integral over all longitudes when the fluxes are relatively constant. A similar behaviour was found in the analysis of ERA5 data (Webb, 2025), where it was found that air detrained in the middle atmosphere over the ITCZ was entrained at other longitudes. As the equatorial winds in the middle atmosphere are predominantly easterlies, the most likely area of entrainment is over the Island Continent.

    In the forced run, the flux at the lowest level of the forced region is slightly less than the control, despite the increased SST.

Above this the reduction in convective flux at 525 hPa is much smaller than in the control. The flux is then unchanged up to 322 hPa. The results imply that the main effects of increased SST is to increase the fraction of deep atmospheric convection and reduce the amount of detrainment in the middle atmosphere. The integrals over all longitudes remain roughly constant with height, close to the values of the control run, implying that the reduced detrainment over the forced region is being is matched by reduced entrainment elsewhere.

Again, this is similar to the behaviour seen in the ERA5 data (Webb, 2025) when, during the development of strong El Niños, increased temperatures along the line of the NECC resulted in increased deep convection in the ITCZ and a reduced entrainment at other longitudes. As ITCZ convection makes a significant contribution to the Hadley Cell, the results imply that increased NECC temperatures may have both a local effect, discussed in the next section, and be responsible for modifying the longitudinal structure of the Hadley Cell

---

[1]See also Appendix B



**Figure 2.** Averaged sea level pressure in September of year 5 for (a) the control run (b) the forced run, both with contour interval of 0.5 hPa, and (c) the change due to the forcing, with contour interval of 0.25 hPa.

### 3.1.2 ITCZ Convection and the Vertical Modes

Although the small reduction in convection during the forced run was unexpected, the increase in deep atmospheric convection can have a local effect on both the local atmospheric pressure field and the wind. Salby and Garcia (1987) found that convection near the Equator excites modes with a vertical scale about twice the height of the convective region. De-Leon et al. (2020) also reported that tropical waves, forced by convective maxima in the mid-troposphere, project strongly onto low vertical modes, the ones with the fastest phase speeds and the largest horizontal scales.

As a result, the increased deep atmospheric convection in the forced run should trigger more of the lowest order atmospheric modes which, because of their scale, will have a greater horizontal impact. As the scale length of the lowest mode is expected to





be around 6 degrees and the ITCZ lies a similar distance from the Equator, it is possible that sea level pressures at the Equator will be affected by the increased deep convection along the line of the ITCZ. This possibility is considered next.

## 3.2 Atmospheric Pressure

Figure 2 shows the average surface pressure in the control and forced runs, together with the difference. Near the Equator zonal pressure gradients in both the control and forced runs are very weak, with a difference of around 4 hPa over 180 degrees of longitude (~20,000 km). Meridionally the gradients are also small, with a difference of around 2 hPa between the Equator and the ITCZ trough at 10° N.

On the Equator, the main effect of the forcing is to slightly reduce the pressure gradient. Figure 2c indicates that this is primarily due to a region of reduced pressure in the east which spreads out from a minimum on the southern boundary near 255° E (105° W), and increased pressures in the west spreading out from maxima on the northern and southern boundaries near 170° E.

In between there is a weaker region of reduced pressure surrounding the forcing region. Within the forcing region itself, this forms a narrow valley with a depth of order 0.02 hPa. Around this there is a broader valley region of similar depth that, between 180° E and 240° E (120° W), extends from approximately 6° S to 11° N. Both valley features are consistent with the expected changes from increased deep convection, discussed earlier.

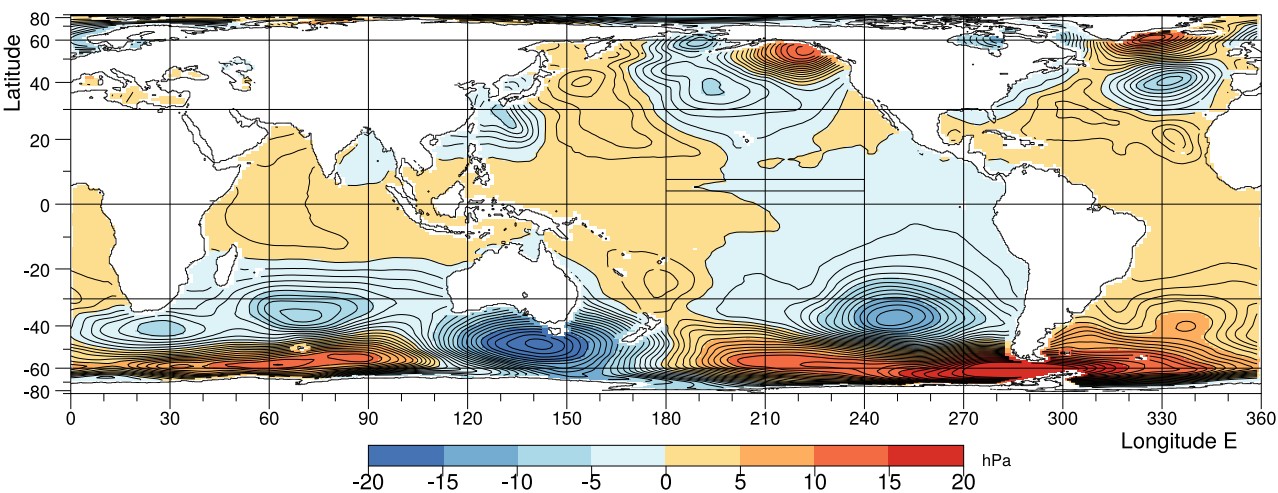

**Figure 3.** Global change in sea level pressure, due to the forcing, averaged over September of year 5. Contour interval 1 hPa.

However, it is the large-scale response which dominates the change in pressure gradient, the result of a drop of around 0.75 hPa in the east and an increase of 0.3 hPa near the dateline. One way the increased pressures in the west could be connected to the increased ITCZ convection is if the maxima north and south of the Equator correspond to the Rossby wave parts of the solution described by Gill (1980). In his solution the maximum increase in surface pressure occurs to the north-west



**Figure 4.** Cloud fraction between the 50 hPa and 300 hPa levels in (a) the control run, (b), the forced run and (c) the difference between the forced and control runs, averaged over September of year 5.





of the forcing region at approximately two non-dimensional scale distances from the Equator. If the maxima are due to Rossby waves generated in this manner, then a larger forcing should result in easterly winds in the western equatorial Pacific, as is observed during El Niño events.

Unfortunately the scale distances are much larger than the value of around 7 degrees estimated from Eqn. 1. In the south, the centre of the pressure maximum (Fig. 3) is near 25° S, corresponding to an equatorial scale of over 12 degrees or a wave speed of over $80\ \mathrm{m\,s^{-1}}$. In the north there, is no similar maximum until 40° N. Thus, although both features may be related to Rossby waves, the evidence is not conclusive.

There is also a problem in the south-east Pacific where the minimum in the pressure is not predicted by theories of either
equatorial deep convection (Matsuno, 1976; Gill, 1980; Vallis, 2017) or the Hadley Cell (Held and Hou, 1980; Schneider, 2006; Hoskins et al., 2020). This is discussed next.

### 3.3   The Large-Scale Response

Figure 3 is a global version of the figure showing the change in atmospheric pressure due to the forcing. Although the forcing is only of order 1° C and is over a limited area, it has generated pressure changes of more than $5\ \mathrm{hPa}$ over large areas of ocean.
In the South Pacific, the mean pressure difference between 30° S and the Equator is generally less than $15\ \mathrm{hPa}$, so the change is large.

During the northern summer, the high pressure region in the south-east Pacific is one of the main Hadley Cell sinking regions. As summer ends, pressures in the region might normally drop, but here the forced run is showing an extra reduction, which continues in the following months.

In retrospect it is possible that such a strong response should have been expected. The earlier studies of Webb (2018b, 2025) showed that El Niños were associated with both a warm NECC, a reduction in deep atmospheric convection over the Island Continent and an increase over the ITCZ.

As both regions are important contributors to the Hadley Cell, it seems inevitable that a change in the convection longitudes, as seen in the forced run, will affect the strength of the descending branches of the cell. As discussed later, global plots of the
vertical flux in the atmosphere show that sinking in the south-east Pacific is reduced in the forced run.

### 3.3.1   Cloudiness

One reason that sinking may be reduced can be seen in Fig. 4 which shows the cloudiness near the top of the troposphere in the control and forced run. North of the Equator, adjacent to the forcing region, cloudiness is increased, the result of the enhanced deep convection in the ITCZ.

However there is also a significant increase of high cloud in the South-eastern Pacific. This is usually a major sinking region and a major part of the Hadley Circulation. The is partly because the region usually lacks high cloud cover. When this occurs it allows a radiative loss of heat to space and results in the increase density of the whole of the upper and middle atmosphere.

If significant amounts of high cloud develop, as occurs here, cooling is reduced, the sinking rate is reduced and with less air reaching the surface, surface atmospheric pressure drops.





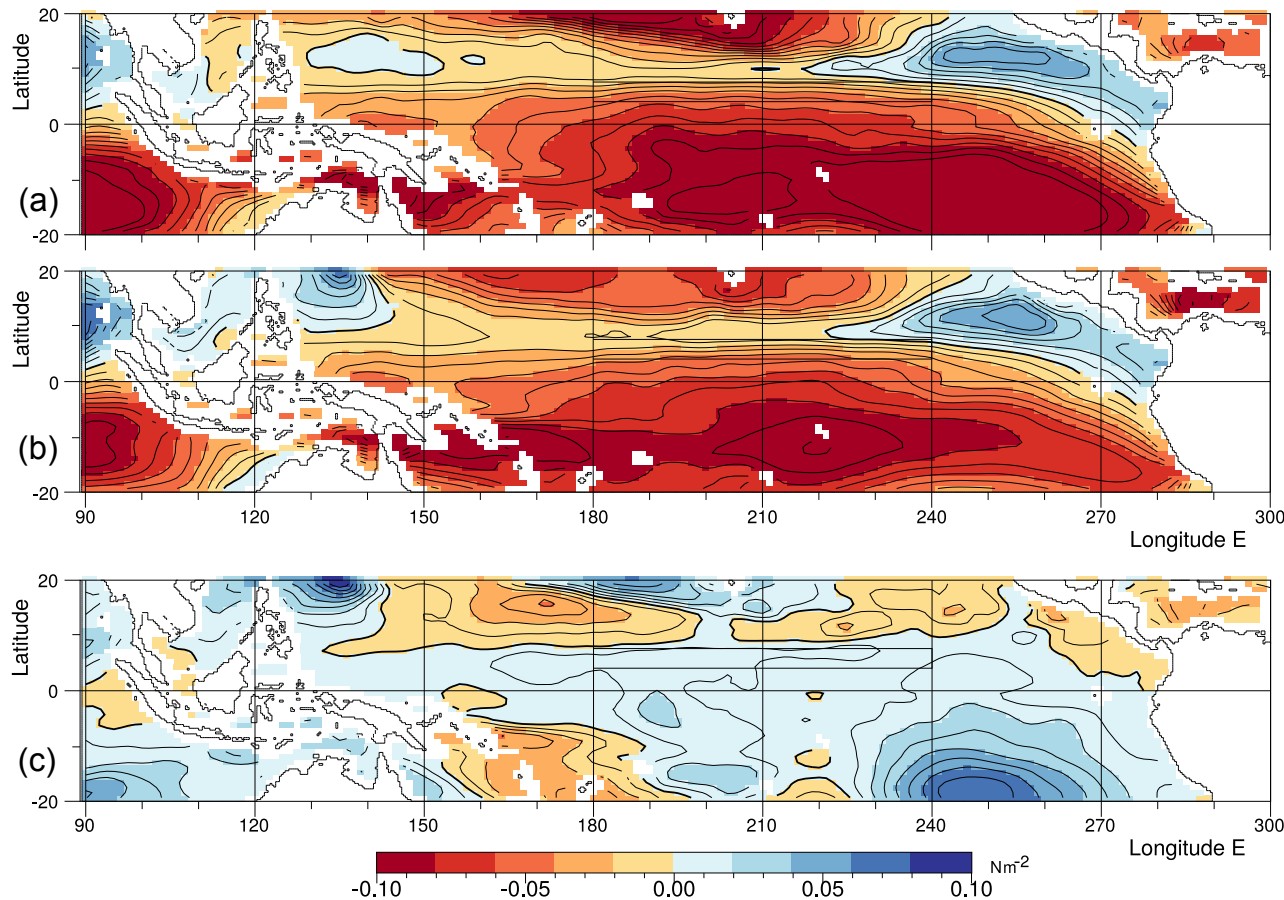

**Figure 5.** Averaged zonal wind stress acting on the ocean in September of year 5 for (a) the control run, (b) the forced run, both with contour interval 0.01 N m$^{-2}$ and (c) the difference, with interval 0.05 N m$^{-2}$.

So if this increased cloudiness is not just a random event, it can help explain why increased deep convection along the line of the ITCZ results in reduced surface pressures in the South Pacific. As discussed in the next section this can have a significant effect on the ocean surface winds.

### 3.3.2 The Southern Oscillation

However, before leaving the global scales, it is noticeable that the pattern of surface pressure change in the Pacific, due to the
forcing, is similar to but of opposite sign to the Southern Oscillation correlation pattern for the autumn months, shown in Fig. 3 of Trenberth and Caron (2000).





This is consistent with the El Niño index being of opposite sign to the Southern Oscillation Index (SOI). The surface pressure field shows reduced pressures, and the SOI correlation field shows positive correlations in a band extending from the SE Pacific to the North Pacific near 190° E (170° W), 40° N. Both patterns have a change of sign near the dateline and values, again of opposite sign, above the Island Continent and Indian Ocean.

## 3.4 Zonal Wind Stress acting on the Ocean

Figure 5 shows the zonal wind stress for the control and forced runs, in September of year 5, together with the difference between the two runs.

The stress acting on the ocean is primarily a function of the surface wind, with positive and negative zonal stresses corresponding to westerly and easterly surface winds. The magnitude of the stress also depends on ocean currents, the surface wave field and near surface turbulence and stratification in both the ocean and atmosphere, but usually such effects are small.

In both runs of the model, the equatorial Pacific is primarily a region of easterly winds forcing the ocean westwards. The easterlies are an extension of the south-east trades, part of the circulation around the high pressures of the South Pacific. In the north, the zonal wind stress is also westwards, due to the north-east trades associated with the high pressures of the North Pacific.

In between is the band associated with the ITCZ. In the central Pacific, this is a region of weak westward wind stress but in the east and west, the wind direction changes. This is most noticeable off Central America where the zonal wind stress is strongly positive.

In the control run, the stress on the Equator in mid-Pacific is typically $-0.05\,\mathrm{N\,m^{-2}}$. This a reasonably large value, the maximum zonal wind stress in the South Pacific between 30° S and the Equator being around $-0.1\,\mathrm{N\,m^{-2}}$.

In the forced run the zonal stress on the Equator in mid-Pacific drops to around $-0.04\,\mathrm{N\,m^{-2}}$. The reduction is small but not insignificant, given that this is a perturbation experiment.

The changes are clearer in Fig. 5c, showing the difference between the two runs. There is a broad band surrounding the Equator, where the response is positive, indicating weaker easterlies in the forced run.

Comparison with Fig. 2 indicates that the reduced wind stresses along the Equator are the result of three effects. In the east, higher equatorial pressures are reduced because of the reduced strength of the sinking region in the South Pacific. This reduces the strength of the south-east trade winds which are linked to the easterlies along the Equator.

In the central Pacific, the direct effect of increased convection along the ITCZ reduces pressure slightly and so helps to reduce pressure gradient and the easterlies.

In the west the low initial pressures are raised slightly due to increased pressures in the western Pacific, north and south of the Equator. Again, this may be a reflection of a changed Southern Oscillation. This pressure increase in the west reduces the pressure drop across the Pacific, and so reduces the wind stress acting on the ocean.

To summarise this section, increasing the temperature of the NECC, has resulted in a significant reduction in the easterly wind stress acting on the equatorial Pacific Ocean. In the central Pacific this may be a direct result of deeper convection within the ITCZ. However, near 170° E and 250° E (110° W), one or more other processes appear to be involved, the drop in surface





stress around 250° E being connected with the changes further south in the Hadley Cell sinking region, and the rise in pressure near 170° E being connected with the possible Rossby wave features in the north-west and south-west Pacific.

**Figure 6.** Ocean temperatures in a section along the Equator averaged during September of year 5 for (a) the control run, (b) the forced run and (c) the temperature change due to the forcing.





## 4 The Response of the Ocean

The above focus on wind stress arises because it is the zonal wind stress close to the Equator that is responsible for many of the

ocean features associated with El Niños and La Niñas. This is partly because the equatorial scale of the atmosphere, discussed earlier, is larger than the 1 to 2 degree scale of the ocean.

On the Equator, the westward wind stress generates an Equatorial Current that carries warm surface water to the west. This results in a deep surface thermocline in the west Pacific, the boundary with the cooler deeper waters sloping upwards from west to east.

The resulting slope in the density surface generates a pressure gradient, which drives an Equatorial Undercurrent. The current lies at a depth of only a few hundred metres and brings cool water from deep layers in the west Pacific to much shallow depths in the east.

Within a degree on either side of the Equator the Coriolis force starts being important and, where the wind stress is towards the west, results in an Ekman transport carrying surface water away from the Equator. The surface water is replaced by cool

water from below, the final temperature of the water reaching the surface being a balance between Ekman driven upwelling and the downward diffusion of heat (Stommel, 1960; Webb, 2018a). Together the processes are responsible for the cold pool of the eastern equatorial Pacific.

During El Niños, the westward wind stress near the Equator is reduced. This results in a smaller thermocline slope, a reduced undercurrent, less upwelling of cold undercurrent water and, as the surface heating remains roughly constant, a general warming

of the cold pool.

In the west, the wind stress may change direction during El Niños. Ekman transport then converges warm surface water in the Equator, where the easterly surface current then carries it to a region of weak winds where deep atmospheric convection develops.

The previous section has shown that the equatorial wind stress is reduced during the forced run. The next two sections

concentrate on whether this is sufficient to generate significant changes to both the ocean temperature structure and the zonal currents on the Equator. The following section concentrates on the resulting changes in the ocean surface temperature field.

### 4.1 The Ocean Thermal Structure at the Equator

Figure 6 shows ocean potential temperature on the Equator in the control and forced runs, together with the difference. The top two figures shows the pool of warm water in the western Pacific, and the shallow thermocline sloping up from depths of

around 200 m in the west to depths close to the surface in the east.

The results are similar to the sections for August and October published by Johnson et al. (2002) and based on ten years of observations. At 160° E, both sets of figures show the thermocline at similar depths and have similar temperature profiles nearer the surface. Both also show the thermocline shallowing towards east with the 23° C contour nearing the ocean surface around 240° E (120° W). However, the mixed layer is much thicker in the model than in the observations.





**Figure 7.** Ocean zonal velocities in a section along the Equator averaged during September of year 5 for (a) the control run, (b) the forced run and (c) the difference.



Johnson et al. (2002) also show the change at each phase of the ENSO cycle, but without specifying the months involved. A comparison of the El Niño and La Niña phases show the surface thermocline shallowing, by about 10 m in the west, but becoming up to 40 m deeper in the central and eastern Pacific. In the western Pacific this results in a slight cooling of the surface layer. In the central and east surface temperatures increase by 2° C or more.

The present perturbation run is not expected to show large changes in thermocline depths and temperatures, but from the
difference field (Fig 6c), near surface temperatures are increased by up to one degree in the central and east Pacific. There is also a slight warming in the west.

At the level of the near-surface thermocline, there is a slight cooling in the west and a significant region of warming in the central and eastern Pacific. This warming is consistent with the reduced upwelling expected resulting from the reduced strength of the easterlies discussed earlier. The reduced upwelling changes the balance between downward diffusion of heat from the
surface and the upwelling of cold water from depth.

## 4.2    Zonal Velocities at the Equator

Figure 7 shows ocean zonal velocity on the Equator in the control and forced runs, together with the difference. This time the agreement with the observations of Johnson et al. (2002) is poor. Both the control run of the model and the observations show a shallow surface Equatorial Current flowing westwards, and a deeper eastward flowing Equatorial Undercurrent. However, the
observations show the the surface current reversing west of the dateline in both August and October. The reversal is not seen in the model.

At the depths of the undercurrent, the observations also show an undercurrent maximum near 155° E that is not seen in the control run. To the east, the observed velocities first reduce slightly before increasing, as in the model, to a maximum around 220° E (140° W).
In the Fig. 7 difference field, the response of the ocean appears to be split into two regions, with a boundary around 190° E (170° W). To the east, where the control run agrees best with the observations, there is a reduction in the strength of the Equatorial Current near the surface, and an undercurrent that moves deeper. The latter produces the band of negative velocity differences near 100 m and the wider band of increased velocities below. The reduction in strength of the westward surface current and the eastward agree with Johnson et al. (2002), but the observations for an El Niño year do not show the increased
velocity at 150m and below seen in the forced run.

In the west, the forced run results in a larger reduction in the speed of the westward flowing surface current. There is also a reduction in the speed of the eastward flowing undercurrent. Although the model does not appear to represent reality so well on the eastern side of the equatorial Pacific, both sets of changes are consistent with, but smaller than, the differences between the Johnson et al. (2002) El Niño and La Niña cases.
To summarise sections 4.1 and 4.2, there are noticeable differences between the control run of the model and the observed ocean, but the near-surface changes in the forced run are consistent with the changes that might be expected during a period of increasing El Niño index.





**Figure 8.** Averaged sea surface temperature in September of year 5 for (a) the control run, (b) the forced run and (c) the difference.

## 4.3 The Surface Temperature Field

The reduced easterlies along the Equator should result in reduced Ekman upwelling and so result in increased temperatures in
the surface layers of the ocean. Figure 8 shows the temperatures in the control and forced runs at the surface, together with
their difference. Surface temperatures along the Equator span the range 20° C to 30° C, a contrast with the few degrees change
in surface temperature that occurs during extreme ENSO events.

A comparison of the large scale temperature field with observations (Levitus, 1982; Rayner et al., 2003) shows that the
model is missing the east Pacific warm pool off Central America and replacing it with ocean temperatures that are too cold.
Surprisingly, Fig. 1 shows convection offshore from Central America in both runs. This is normally an indication of warm
SSTs.





**Figure 9.** Averaged ocean temperature at 125m in September of year 5 for (a) the control run, (b) the forced run and (c) the difference.

The cold ocean temperatures off Central America may be connected to the North Equatorial Current, which is too cold east of 240° E. The warm pool in the western Pacific is also too small and too cold. In marked contrast the temperatures in the South Pacific warm pool are consistent with observations.

In terms of the oceanography these deficiencies are serious. They almost certainly had some effect on the results presented in this paper.

Fig. 8, showing the effect of the perturbation during September, confirms the pattern of warming along the Equator seen in Fig. 6, with weak surface warming in the central Pacific and a stronger response east of 240° E (120° W). It also shows that in the central Pacific region between 180° E and 240° E the source of the heating is the forcing region to the north, the pattern of
advection indicating that the warm water is being advected southward by tropical instability waves.




Between 240° E and 270° E, the surface temperatures are not affected by the forcing region, so the warming both here and in Fig. 6 is almost certainly a result of reduced upwelling. Vertical velocities are discussed further in Appendix A, where the results confirm that there is reduced near-surface upwelling in this region.

## 4.4 Temperatures at 125 m

Figure 9 contains the corresponding plots for a depth of 125 m, the depth where in the central Pacific Fig. 6 showed greatest warming. Although this is only just below the surface mixed layer, the range of temperatures along the Equator is much larger, extending from 12° C in the east to 30° C in the west.

Within the forcing region there is, as expected, a slight increase in temperature, but it also shows a region of strong cooling in the east. The latter is a result of the time independent forcing function, in which the temperatures are fixed to those at the beginning of August. As a result in September, when advection by the NECC carries warm water into the eastern part of the zone, the forcing function acts to cool the surface waters in this region.

Elsewhere there is a cooling within a band just north of the forcing latitudes, and warming near the Equator, roughly coinciding with the forcing longitudes. The cooling to the north could be due Ekman induced upwelling, a result of the increased easterlies just north of the forcing region (see Fig. 5).

The most important response is along the Equator. Here in the central Pacific the warming takes on the triangular shape of cold pool warming which characterises the El Niño signal. The signal disappears in the east where Fig. 6 shows that the surface thermocline lies above 125 m.

## 5 Comparison of Results from Different Years

So far the analysis has concentrated on the results from a single year. Given the almost random nature of much of the climate system, the robustness of the results was tested by using a second long run of the unforced model. An upgrade in the operating system compiler and subroutines meant that it was not possible to exactly replicate the original long run. During the first two years of the second control run the Niño 3.4 index stayed close to the original but it subsequently drifted away.

The second control run was then used to initialize a series of forced experiments. These started the 1st August in each of the first ten years and used a forcing function, dependent on longitude, similar to that of the original experiment but based on the SST field of the control run that year.

Key results are summarised in Figs. 10 to 13. The value calculated from each of the unforced runs are shown in blue and the results from the forced runs, where the average sea surface temperature in the forced region was increased by 1° C, are shown in red. The triangular symbols refer to the original run.

## 5.1 Omega at 800 hPa

Figure 10 is a plot of the average value of omega at 800 hPa over the forcing region in each September of the control runs and in the corresponding forced run. It shows that at temperatures above 28° C, convection over the forcing region is to first order





a linear function of sea surface temperature, the best linear fit to the data being:

$$\Omega_{800} \quad = \quad -0.0314 * (T_{SST} - 28.13). \tag{8}$$

where $T_{SST}$ is the average sea surface temperature.

The previous studies of Gadgil et al. (1984), Evans and Webster (2014) and Kubar and Jiang (2019) indicated that a critical temperature needed to be passed before deep atmospheric convection developed over the tropical ocean, but the model results imply that the same may be true at low levels of the atmosphere.

The plot also shows that in some years, convection in the control run was unusually low. On inspection it was found that these corresponded to months during which the ITCZ lay further north than usual at the longitudes of the forcing region.

## 5.2 Omega at 300 hPa

Figure 11 is the corresponding plot at 300 hPa. Again the results indicate that to first order convection is a almost linear function of temperature. The linear fit to the data is now,

$$\Omega_{300} \quad = \quad -0.0405 * (T_{SST} - 28.89). \tag{9}$$

The results indicates that higher temperatures are needed for convection to reach 300 hPa and that the increase of flux with temperature is about a third greater than at 800 hPa.

Webb (2025) found that under normal conditions most of the convective flux crossing the 800 hPa level in the ITCZ failed to reach the 300 hPa level. Instead it appeared to be advected westwards where it could become entrained by the convective plumes over the over the Island Continent.

The linear relationships of Eqns. 8 and 9, imply that, above the forcing region, detrainment in the middle atmosphere increases with SST between 28.1° C and 28.9°C and that, at higher SST values, detrainment will decrease with temperature, becoming zero around 30.5°C. However it is possible that at higher temperatures non-linearities are important, Figs 10 and 11 indicating that above 30°C, detrainment has usually stopped and there is a net entrainment by convection.

### 5.3 Surface Pressure in the SE Pacific

In the discussion of the first run analysed, three mechanisms were discussed that might influence the pressure gradient along the Equator. The first was the direct result of the increased height of deep convection. The second was a possible Rossby wave response in the western Pacific and the third was the possible impact of increased ITCZ convection on pressure in the south-east Pacific.

Fig. 3 shows that the third region may also be affected by the much larger changes in pressure that occur further south, in the storm belts south of 40° S. This is an area of strong winds and deep depressions, so changes here are almost certainly due to the growth of random meteorological noise. There is thus the possibility that the reduction in surface pressure, observed further north in the forced run, is also due to noise and is not to the direct effect of the forcing.

This possibility was tested by calculating the average surface pressure in each of the model months over the region between 240° E to 270° E (120° W to 90° W) and 30° S to 10° S. These are plotted in Fig. 12.





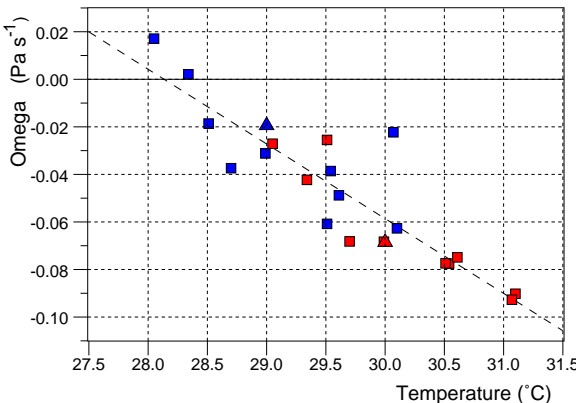

**Figure 10.** Average values of Omega over the forcing region at 800 hPa in each September, plotted against the average sea surface temperature in the region. Blue symbols correspond to the control runs, red to the forced runs. The triangles are from the original run, the squares from the second set of runs. The dotted line is a linear fit to the data from both the control and forced runs.

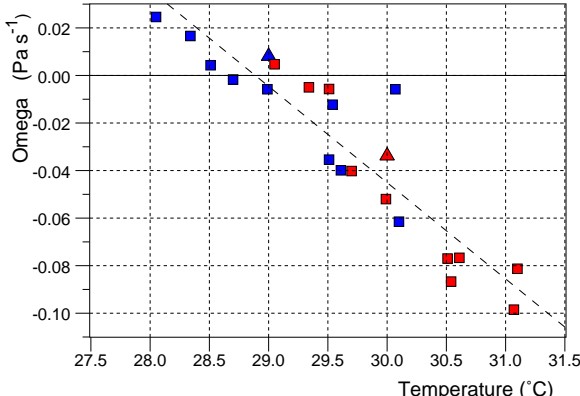

**Figure 11.** Average values of Omega over the forcing region at 300 hPa in each September, plotted against the average sea surface temperature in the region. Blue symbols correspond to the control runs, red to the forced runs. The triangles are from the original run, the squares from the second set of runs. The dotted line is a linear fit to the data from both the control and forced runs.

The results show a large amount of scatter, indicating that random meteorological noise is an important factor. However
despite the noise, there is a tendency for the surface pressure in the south-east Pacific to decrease with increased SST in the forcing region, the change being of order 2 hPa for each one degree increase in SST.



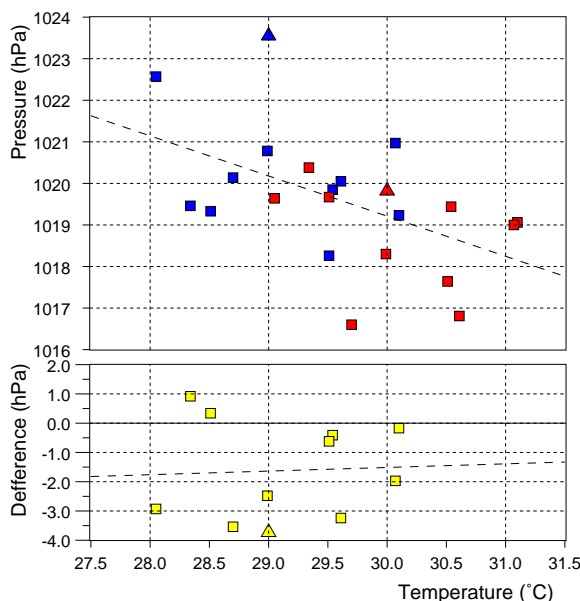

**Figure 12.** (Top) Surface pressures in the SE Pacific plotted against the average temperature in the forcing region. Blue symbols correspond to the control runs, red to the forced runs. Triangles correspond to the original run. (Bottom) Differences in the pressures plotted against temperature of the control run.

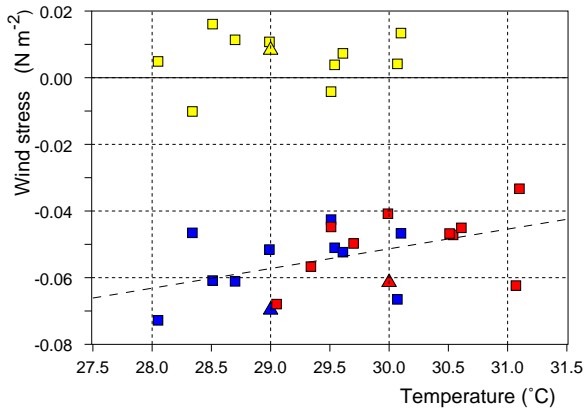

**Figure 13.** Average values of the zonal wind stress on the Equator between 180° E and 240° E, plotted against the average temperature in the forcing region. Stress in units of $\mathrm{N\,m^{-2}}$. Blue symbols correspond to the control runs, red to the forced runs. Triangles correspond to the original run. Yellow symbols indicate the difference between the control and forced runs, plotted against the temperature of the control run. The average change in stress is $0.0057\ \mathrm{N\,m^{-2}}$.



## 5.4 Wind Stress Along the Equator

Figure 13 shows the average zonal wind stress acting on the equatorial ocean between 180° E and 240° E (120° W). There is again a significant amount of meteorological noise but the linear fit to the data from both the control and forced runs shows that the zonal stress increases by 0.0059 $\mathrm{Nm}^{-2}$ for each one degree increase in temperature.

The figure also shows the difference in average wind stress in each pair of control and forced runs. In two case the westward wind stress increased, but in the rest, the wind stress declined. The average value of the pairs is -0.0060 $\mathrm{N\,m}^{-2}$, essentially the same as the linear fit to the full set of data[2].

## 5.5 Comparison with ERA5 Fields

Other sources of information about the fields being discussed is limited, and so to provide some context, appendix B includes data from the ERA5 reanalysis during the period 1979 to 2021. As seen in Fig. B1, the ERA5 temperatures in the forcing region, based on observations, span a range from 27.3° C to 29.4° C. The difference between the minimum and maximum values is similar to that of the CESM control runs but the ERA5 temperatures are about 0.5° C cooler.

Figure B2 shows the average vertical flux over the forcing region at 800 $\mathrm{hPa}$ based on the ERA5 data. The estimates range from -5 $\mathrm{Tgs}^{-1}$ to -17 $\mathrm{Tgs}^{-1}$, corresponding to values of Omega between -0.027 $\mathrm{Pas}^{-1}$ and -0.090 $\mathrm{Pas}^{-1}$. Comparison with Fig. 10 shows that the range of values is similar to that of the CESM runs but that convection in the forcing region is, in the ERA5 reanalysis, more intense than in the CESM runs. This is despite the fact that the ERA5 estimates of sea surface temperature are lower.

Figure B3 shows Omega at 300 $\mathrm{hPa}$. In the forcing region, the ERA5 mass flux usually lies between zero and -10 $\mathrm{Tgs}^{-1}$ (-0.053 $\mathrm{Pas}^{-1}$), but the figure shows that during the development of the 1997-98 and 2015-16 El Niños it reached -15 $\mathrm{Tgs}^{-1}$ (-0.079 $\mathrm{Pas}^{-1}$).

Comparison with Fig. 11 shows that the range of values, 0.08 $\mathrm{Pas}^{-1}$, is similar in both datasets but that within the forcing region the CESM model tends to have more sinking air and less convection. However whereas the CESM convection appears to be roughly a function of SST, in the ERA data there are two years where the flux is almost 50% greater than the maximum in other years. This occurs despite the sea surface temperature being similar to or less than 0.5° C warmer than at the time of the other maxima.

Figure B5 shows the ERA5 estimate of the average wind stress along the Equator between 180°E and 240°E. In the CESM runs, the average zonal stress was around -0.06 $\mathrm{Nm}^{-2}$ in reasonable agreement with the ERA5 data in which values are around -0.05 $\mathrm{Nm}^{-2}$. The CESM tests showed a small correlation with increased temperature forcing region and this is also seen in the ERA5 data, with some of the extreme values occurring during the years when strong El Niños were developing.

Although the zonal stress on the equator continued to force the surface westwards westward during these years, the ERA5 reanalysis plotted in Fig. B4 shows that the zonal pressure gradient was close to zero. The figure also shows the pressure

---

[2]The plot of the surface pressure difference on the Equator between 240° E (120° W) and 180° E is similar. Differences are around 3.5 hPa at 28° C, and decrease to 2 hPa at 30° C. The drop in each of the forced runs is around 0.5 hPa.





difference between Tahiti and Darwin, used as a measure of the Southern Oscillation, and the difference between the south-
east Pacific and Darwin. The CESM runs indicate that a temperature increases of a degree in the forcing region could reduce
pressure in the southeast Pacific by a few hPa. The results from the ERA5 analysis are largely consistent with this.

## 6   Summary

### 6.1   Main Results

The study reports on a set of perturbation experiments, using the CESM climate model, in which in eleven different years, a
forced run was started on the 1st August. in which temperatures along the path of the North Equatorial Counter Current were
increased by one degree Centigrade. The model fields, averaged over the following month, were then analysed to determine
what systematic changes had occurred due to the forcing.

   The results showed at sea surface temperatures (SSTs) above 28.1°C there was a linear increase with SST in the convection
through the 800 hPa level in the atmosphere and that above 28.9°there was a faster increase with SST in convection through
the 300 hPa level. As the ITCZ region is one of the major regions involved in deep atmospheric convection, the increased flux
at 300 hPa is likely to have a significant effect on the zonal structure of the Hadley Circulation, and so have a world-wide
impact.

   The results also showed changes near sea level which could feed back into the ocean. On the Equator the horizontal pressure
gradient between 240° E (120° W) and 180° W fell, by an amount comparable with the observed year to year fluctuations, and
this resulted in comparable changes in the wind stress acting on the ocean.

The results provided evidence that the reduction in pressure gradient was, in part, directly due to increased convection over
the warmed ocean. The increased convection itself will reduce sea level pressures along the line of convection. In addition the
greater vertical scale is expected to extend the horizontal scale of the region of low pressure.

   Unexpectedly the results also showed a decrease of surface pressures in the south-east Pacific. As the trades winds in the
South Pacific are a response to the sinking air and resulting high pressures in the region, and as these trades also extend to the
cross the Equator, the reduction in pressure also acts to reduce the westward wind stress acting on the ocean along the Equator.

   One of the main features of the equatorial Pacific Ocean, is the Equatorial Cold Pool, a region of low SSTs cause by the
easterly winds near the Equator, together with the Coriolis force, forcing the surface layers to north and south, so that it is
replaced from cold water from below.

   As a result of this connection, the reduction in easterly wind stress observed in the forced runs, should result in reduced
upwelling and some warming of the near surface layer. In the perturbation runs the warming was greatest around 125 m, near
the bottom of the surface mixed layer where the upwelling velocities have a maximum. Changes were also observed in the near
surface Equatorial Current and the deeper Equatorial Undercurrent.

   Overall the perturbation runs reproduced many of the fluctuations usually associated with the normal pattern of the El
Niño/La Niña signal. However they did not produced any marked change in the surface temperatures in the Cold Pool region.
There were changes in the winds along the Equator, but they were not enough to result in westerly winds in the western Pacific





associated with El Niños. They also did not result in the development of a deep convection region on the Equator near the dateline, which is also associated with El Niños.

## 6.2 Concerns

The model sea surface temperature field in the North Pacific is anomalous. The east Pacific warm pool off central America is
missing and the west Pacific warm pool is cool and of restricted size.

This is unfortunate and needs to be addressed either by correcting the CMIP model or by using another model with a better representation of the SST field. There is however no evidence that the error has affected the main results of this paper, which concern changes occurring nearer the Equator and south of the Equator.

## 6.3 Mechanisms

It was initially expected that increased convection along the line of the ITCZ would reduce sea surface pressure locally and, by exciting atmospheric modes with a large horizontal scale, would affect surface pressures and winds on the Equator. The results supported this theory, the increased deep convection reducing pressures along the line of convection and also extending its area of influence towards the equator.

The results also showed that one or more other mechanisms were involved. Webb (2025) found, in the ERA5 reanalysis,
that during the development of strong El Niños, increased convection along the line of the ITCZ affected the longitudinal structure of the rising branch of the Hadley Cell. Something similar occurs in the present forced runs, the longitude structure of the Hadley Cell changes but the the total strength of the cell is only slightly affected by the increase in deep atmospheric convection within the ITCZ.

This increased ITCZ convection may be responsible for the high surface pressures seen in Fig. 2 around 170° E, via a Rossby
wave mechanism similar to the ones that develop in the Gill (1980) study of convection near the Equator. It is also noticeable that the pressure signal associated with the Southern Oscillation has similarities with the increase in pressure seen in the forced run, and may help explain the observed correlation between El Niños and the Southern Oscillation.

The longitudinal change in Hadley Cell convection may be responsible for the low sea surface pressures that develop in the South Pacific, via its effect on the Hadley Cell sinking regions. The present results show that although surface pressures are
reduced by the forcing, the response is also affected by a large amount of meteorological noise. This indicates that Hadley Cell sinking is only weakly dependent on the distribution of convection and may instead depend primarily on variations in radiative cooling due, for example, to the distribution of cloudiness, discussed earlier, and fluctuations in humidity and the concentration of particulates. Such factors might also be affected by processes such as the Madden-Julian oscillation.

## 6.4 Hypothesis

The original hypothesis was "that the changes observed during El Niños are, at least in part, a response of the climate system to changes in sea surface temperature along the path of the North Equatorial Counter Current". It is possible that the term





"El Niños" should be replaced by "ENSO" but whatever is chosen, in physical systems such statements can never be proved correct.

The present set of perturbation tests, using a respected climate model, have shown El Niño type changes in the equatorial
Pacific connected to major changes in convection over the ITCZ and the structure of the Hadley Cell. In terms of the hypothesis, the central Pacific response means that the hypothesis has not been proved to be wrong. This is encouraging but, possibly more usefully, the study has also suggested a set of physical mechanisms which support the hypothesis.

The forcing did not result in surface warming of the Pacific Cold Pool, but reduced upwelling did occur, so this may be due to the short period studied. The forcing also did not result in increased convection on the Equator near the dateline or increased
westerlies in the western Pacific. Here there is no mitigating feature so, in these cases, other factors, possibly the reduced temperature of the warm pool, must also be involved.

## 6.5 Finally

In conclusion, if the results presented here are valid, then fluctuations in the temperature of the North Equatorial Counter Current are likely to contribute to the normal fluctuations of El Niño and La Niña in the central Pacific, and especially to the
events associated with the strong El Niños of 1983-84, 1997-98 and 2015-16.




**Appendix A: The Vertical Velocity Field**

The main paper shows that the perturbation of sea surface temperature along the path of the North Equatorial Counter Current (NECC) resulted in a reduction in the strength the atmospheric easterlies along part of the Equator and a warming of the near-surface ocean, with a maximum at depths near 100 m. Stommel (1960) showed that the easterlies together with resulting Ekman Transport close to the Equator, generate both the surface Equatorial Current and the shallow Tropical Cell in which water is upwelled on and near the Equator and sinks a few degrees to the north and south.

In the region of upwelling, the near-surface vertical temperature structure in the ocean is dominated by a balance between the upwelled cold water and the downward transport of heat by mixing. As a result any reduction in the flux of upwelled water at some depth results in an increase of temperature at that depth. To a lesser extent, the temperature can also be affected by changes in the temperature of water advected from the east by the Equatorial Current and from the west by the deeper Equatorial Undercurrent.

Figure 6, in the main paper, shows that in the central Pacific, the largest temperature changes due to the forcing occur at a depth of around 125 m. The depth of the maximum becomes shallower in the eastern Pacific, reaching the surface around 260° E (100° W).

Figure A1 shows the average vertical velocities at a depth of 125 m during September of year 5. Velocities are of order of $1\,\mathrm{cm\,s^{-1}}$, corresponding to $100\,\mathrm{m\,day^{-1}}$. In the control run there is a band of upwelling water along the Equator extending one to two degrees north and south. This is strongest in the western and central Pacific, and in these regions there matching bands of downwelling water lying just to the north and south. In the north the interface between the two bands also shows additional maxima in both upwelling and downwelling. There are also east-west fluctuations in the strength of both upwelling and downwelling bands, at a scale consistent with them being due to the presence of tropical instability waves.

The changes in the forced run are small but there is a weakening in the strength of upwelling near the Equator and a weakening in the corresponding downwelling, most noticeably in the north.

Figure A2 shows the vertical velocities in a section along the Equator. In the control run, the maximum upwelling in region west of 210° E, usually lies at a depth around 80 m. Comparison with Fig. 5 of the main paper shows that this lies near the bottom of the surface mixed layer.

Near 210° E the level of maximum upwelling drops to below 100 m, where it lies closer to the near-surface thermocline. Further east it rises along with the thermocline, reaching 50 m near 260° E.

East of 210° E, the forced run shows a slight reduction in the maximum upwelling velocity. As this is in a region with a large vertical temperature gradient, it helps to explain the increased temperature at this level.

Figures A3 and A4 are plots of averages made between 210° E and 240° E of the vertical and northward components of velocity. Figure A3 shows the expected maximum in vertical velocity on the Equator associated with the Equatorial Cell, in both the control and forced runs, and Fig. A4 shows the expected meridional currents in the southern hemisphere in both runs.

North of the Equator, the forced run shows additional overturning features near 3° N and 4° N. The one at 4° N occurs only in the forced run and can be explained as an artifact of the forcing. The feature at near 3° N is found in both the control and




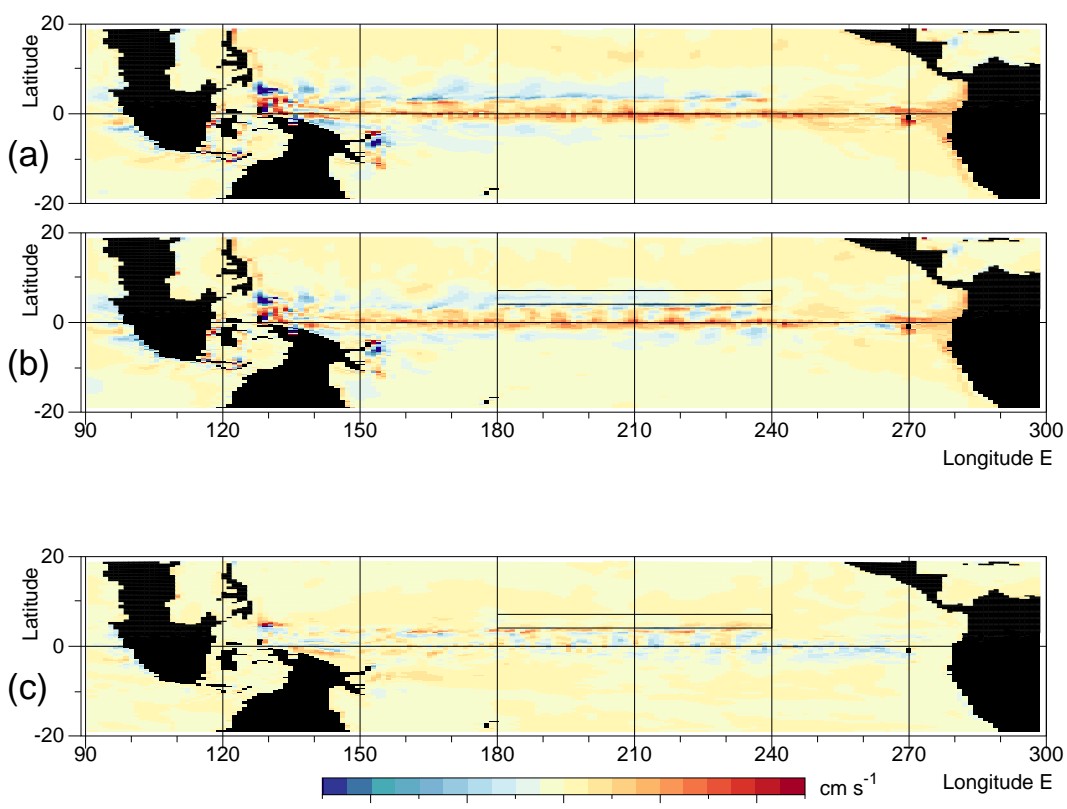

**Figure A1.** Averaged upwelling velocity at a depth of 125 m in September of year 5 for (a) the control run, (b) the forced run (c) the difference.

forced runs. The region is associated with the growth of tropical instability waves[3], so if the water advected north by the eddies is cooled by mixing, and that advected south is warmed, it could explain the extra overturning seen here.

Overall the results imply that the Stommel theory is valid near to and south of the Equator but that north of the Equator, an extra overturning, probably due to tropical instability waves, is also a factor.

## Appendix B: ERA5 Summary Fields

Figures B1 to B5 are based on data from the ECMWF ERA5 atmospheric reanalysis project (Hersbach et al., 2022) and are included here to summarise the changes that occur from year to year. Each figure shows an average field calculated from the monthly averages archived for September during each year between 1979 and 2021. Each figure also shows time series of a key index calculated from the same set of data.

---

[3]Better described as tropical instability eddies




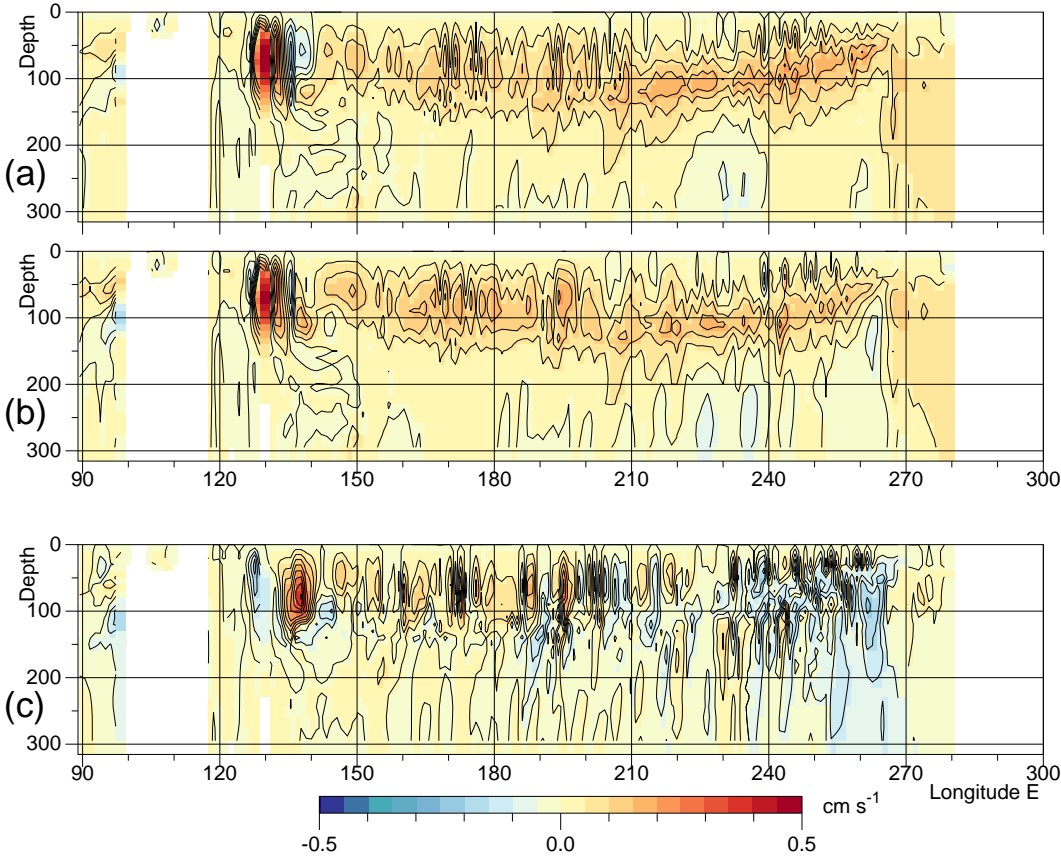

**Figure A2.** Upwelling velocity, averaged between 1° N and 1° S during September of year 5 for (a) the control run, (b) the forced run (c) the difference.

## B1  Sea Surface Temperature

Figure B1 shows the average sea surface temperature calculated from the monthly data together with time series of the average temperature in the forcing region and the El Niño 3.4 index. The two time series are strongly correlated but the temperature changes along the path of the NECC are smaller than those contributing to the El Niño index.

## B2  Convection

Figure B2 shows the Lagrangian rate of pressure change (Omega) at 800 hPa, averaged over each September between 1979 625  and 2021. The top panel, showing the long term average, emphasises the importance of the Intertropical Convergence Zone in the North Pacific. The area of sinking air in the South-eastern Pacific is also a major feature.



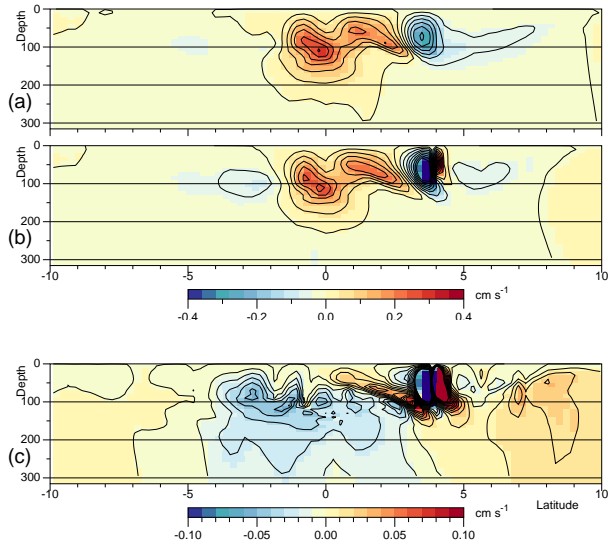

**Figure A3.** Vertical component of velocity, averaged between 210° E and 240° E during September of year 5 for (a) the control run, (b) the forced run (c) the difference.

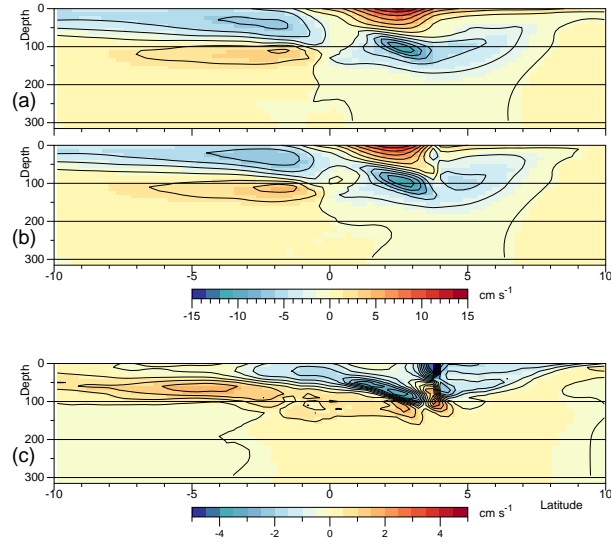

**Figure A4.** Northward component of velocity, averaged between 210° E and 240° E during September of year 5 for (a) the control run, (b) the forced run (c) the difference.





The time series panel shows the total vertical mass flux above the forcing region and the total flux is a band between the same latitude limits but including all longitudes, the latter being offset by $100\,\mathrm{Tg\,s^{-1}}$.

Fluxes in the forcing region average around $10\,\mathrm{Tg\,s^{-1}}$, the values being slightly less before 1988 and slightly more after-wards. The value reaches $18\,\mathrm{Tg\,s^{-1}}$ during the development of the 1998-99 El Niño, but no other El Niño event generates such a large peak. As a result, comparison with the SST time series of Fig B1 shows that, with the exception of 1997-98, the time series have little in common.

Figure B3 shows Omega at $300\,\mathrm{hPa}$, near the top of the troposphere. As discussed in Webb (2025), in a normal year, convection at this level over the ITCZ is much less that at $800\,\mathrm{hPa}$, but convection increases over the Island Continent due to entrainment by the convective plumes.

At $300\,\mathrm{hPa}$ the average flux in the forcing region is reduced to around $5\,\mathrm{Tg\,s^{-1}}$, but the correlations with the Niño 3.4 index is stronger. Omega is particularly large during the development of the 1998-99 and 2015-16 El Niños, indicating that during these times convection was essentially independent of height.

## B3  Surface Pressure

Figure B4 shows the averages of the September sea level pressures during the period 1979 to 2021 and plots of the pressure difference along the Equator between $240°$ E and $180°$ E and that between Darwin and Tahiti. The negative of the pressure between Darwin and Tahiti is convectionally used as an index measuring the strength of the Southern Oscillation, although in his original study Walker (1928) also used observations made further east in the Pacific and further west in the Indian Ocean.

The pressure difference along the Equator is typically around $3\,\mathrm{hPa}$, but this drops to around $2\,\mathrm{hPa}$ when the 1998-89 El Niño is developing and to $1\,\mathrm{hPa}$ or less for the strong El Niños of 1982-83, 1997-98 and 2015-16. The Darwin-Tahiti pressure difference is more variable but it also has strong negative peaks at the time of the 1997-98 and 2015-16 El Niños.

## B4  Zonal Winds

Figure B5 shows shows the eastward component of the near surface ($10\,\mathrm{m}$) winds. The top panel shows that in the equatorial Pacific the winds are predominantly easterlies.

The second panel shows the average of the eastward component on the Equator between $180°$ E and $240°$ E. It illustrates the consistency of the winds on the Equator during most years and the drop in the average winds that occurred during the development of the strong El Niños. This occurred because of the influx of westerlies into the western part of the section.





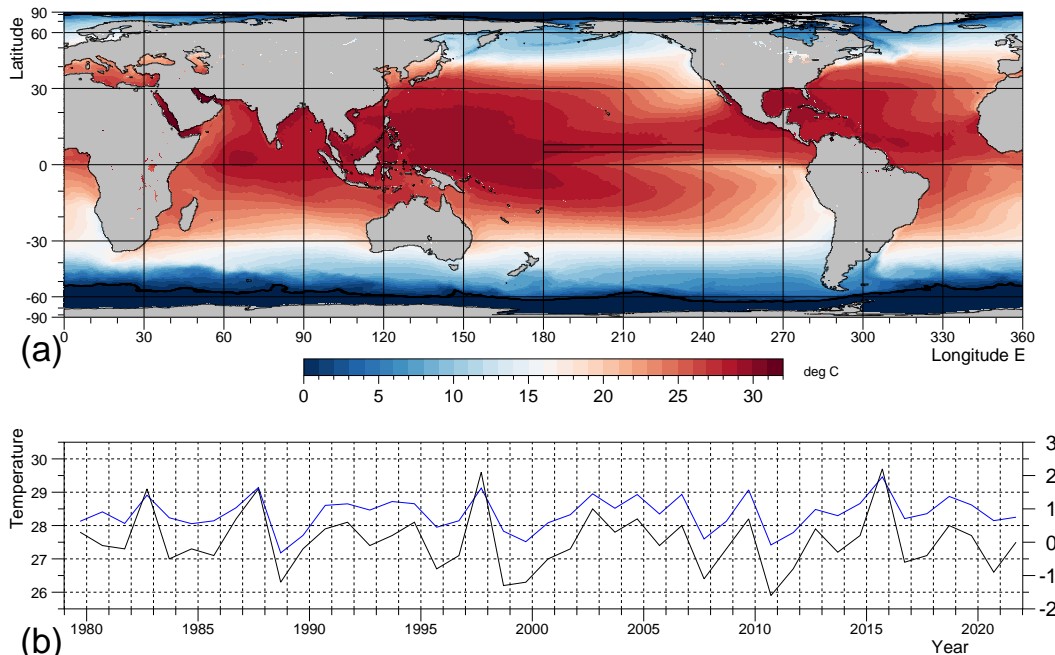

(a)

(b)

**Figure B1.** (a) Sea surface temperature averaged over each September during the period 1979 to 2021. (b) Monthly average for each September within the forcing region (blue) and the El Niño 3.4 index (dashed line). The forcing region lies between 180° E and 240° E (120° W) and between 5° N and 12° N. The El Niño 3.4 index is based on sea surface temperatures between 190° E and 240° E, and between 5° S and 5° N.

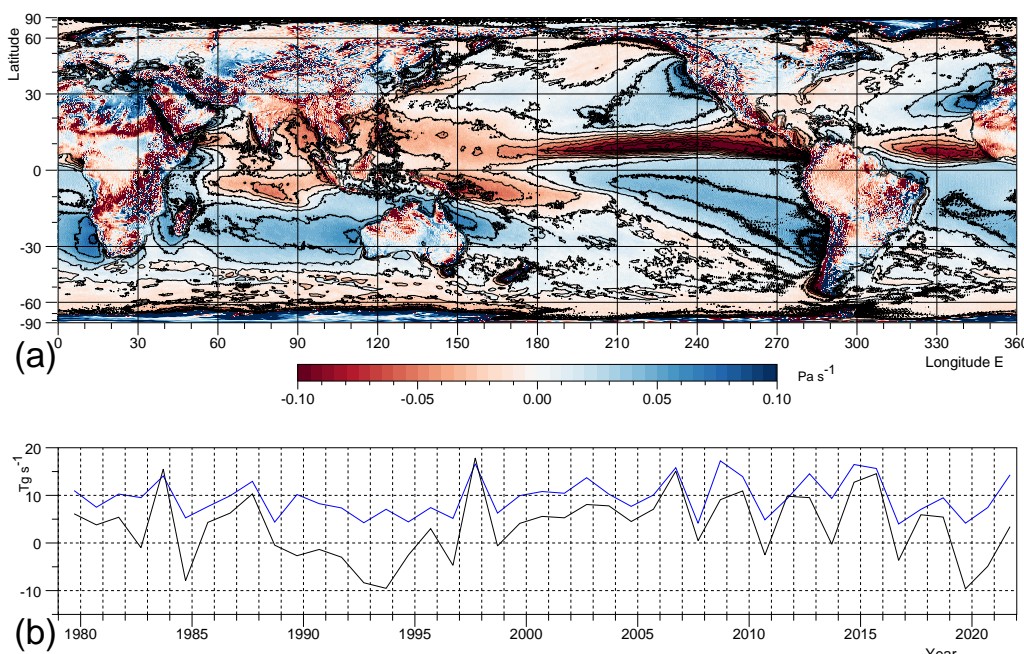

**Figure B2.** (a) Lagrangian rate of change of pressure (Omega) at 800 hPa averaged over each September during the period 1979 to 2021. (b) Total vertical mass flux in the forcing region (blue), and total vertical mass flux over the whole band between 4.75° N and 7.5° N (black), the latter after subtracting 100 $\mathrm{Tg\,s^{-1}}$. Over the forcing region a flux of 10 $\mathrm{Tg\,s^{-1}}$ corresponds to an Omega of 0.053 $\mathrm{Pa\,s^{-1}}$. For the full zonal integral 100 $\mathrm{Tg\,s^{-1}}$ corresponds to 0.088 $\mathrm{Pa\,s^{-1}}$





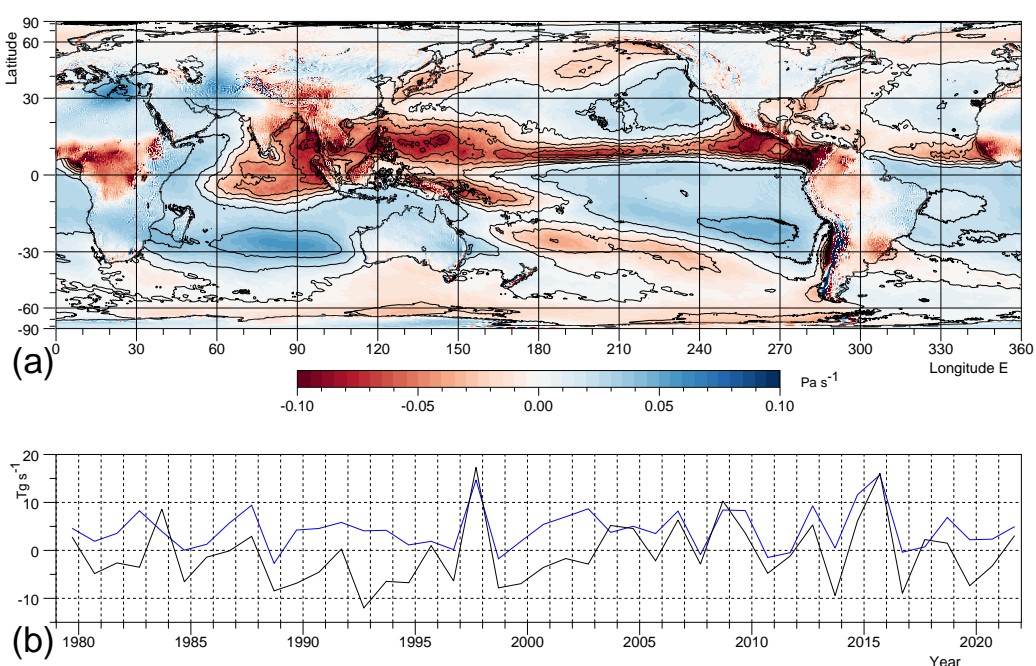

**Figure B3.** (a) Lagrangian rate of change of pressure (Omega) at 300 hPa averaged over each September during the period 1979 to 2021. (b) Total vertical mass flux in the forcing region (blue), and over the whole band (black), the latter after subtracting $100\ \mathrm{Tg\,s^{-1}}$.





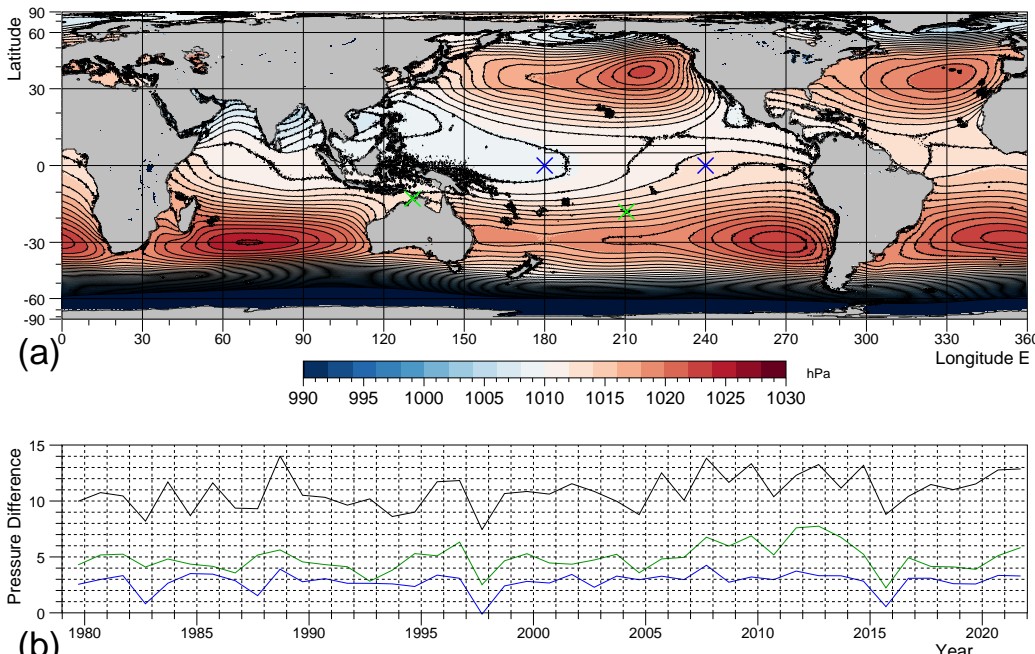

**Figure B4.** (a) ERA5 atmospheric surface pressure averaged over each September during the period 1979 to 2021. Contour interval is 1 hPa. (b) Monthly average for each September of the pressure difference between 240° E and 180° E (blue) and between Tahiti and Darwin (green) during the same period. The locations are shown by coloured crosses. The black line is the average pressure difference between the south west Pacific (240°-270°E, 10°S-30°S) and Darwin.



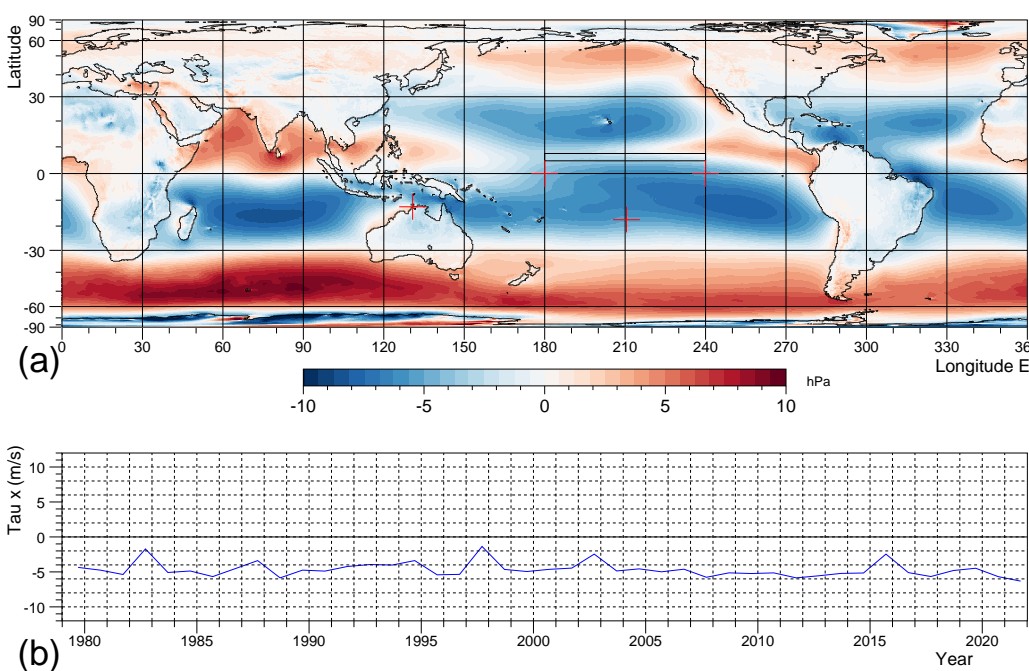

**Figure B5.** (a) ERA5 zonal wind stress acting on the ocean averaged over each September during the period 1979 to 2021. The rectangle shows the forcing region used in the CESM tests. (b) Average along the Equator between 180° E to 240° E for each September during the same period.



*Acknowledgements.* The present study was carried out at the National Oceanography Center, UK, and funded by the Natural Environment Research Council grant NE/Y005589/1. I would like to acknowledge the generous support of NOC and its staff, especially the members of the MSM group.

I would also like to acknowledge previous support, especially that concerned with my interest in the equatorial Pacific, by the UK Institute of Oceanographic Sciences and the CSIRO Division of Fisheries and Oceanography.

The study made use of version 2.1.3 of the CESM2 climate model. The model is made available and supported by the U.S. National Center for Atmospheric Research under the sponsorship of the National Science Foundation. Thanks to all the scientists, software engineers and administrators who contributed to the development of CESM2.

The ERA5 atmospheric reanalysis data was generated by the European Centre for Medium-Range Weather Forecasts and provided for analysis by the Copernicus Climate Change and Atmosphere Monitoring Services (Hersbach et al., 2022).

*Code availability.* The CESM model is available from NCAR. A copy of the modified ocean model subroutine used for the forced run is available from "https://github.com/djwebb/ccode".

*Author contributions.* N/A - single author paper

*Competing interests.* The author was, with Prof. J. Johnson, one of the founding editors of Ocean Science.



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
