# Peer review of "ENSO and the Temperature of the North Equatorial Counter"

_EGUsphere, 2025_

## Referee Comment (RC1)

**Review of "On the Response of the Equatorial Atmosphere and Ocean to Changes in Sea Surface Temperature along the Path of the North Equatorial Counter Current" submitted to EGUsphere by David J. Webb**

**Summary**: An hypothesis, that variations in the SST in the North Equatorial Counter Current (NECC) influence the development of El Nino events, is examined through a set of experiments in which the SST in the mid-Pacific NECC is forced to be 1°C higher than in the control simulation. The atmospheric responses in local convection, surface pressure and zonal surface winds along the equator, and surface pressure in the south-east Pacific are investigated as are the changes in the depth of the thermocline and upwelling on the equator.

**Overall evaluation**: I have previously recommended that an earlier version of this paper be accepted following suitable revisions. The new version of the paper seems to have been appropriately adjusted in response to my suggestions. It also includes a new section reporting results from an ensemble of additional forced integrations which give useful insight into the robustness of the original conclusions. Two interesting appendices have been added and the concluding section improved. My opinion is still that this is an interesting paper that examines an extensively studied and very important subject from an unusual angle and that its discussion of the results and the mechanisms are interesting. The author carefully refrains from unjustified claims about their results.

**Detailed comments**

I provided a lot of comments on the previous version and have limited my comments to the most important points this time.

**Title:** This seems appropriate but it is not very catchy and does not mention El Nino.

**Abstract:** The first two sentences are really good. The next sentence (line 6) might mention all the impacts on the equator, particularly the reduction in the surface pressure gradient and zonal winds and the tilt of the thermocline there. It is probably worth mentioning the impact on SST at the equator; it is increased by at least 0.5°C over 210-270°E (see figure 6). The impacts on the surface pressure in the south-east Pacific are smaller on average in the ensemble experiments than the original experiment (see more detailed discussion of that below). I wonder how the 2 mb change compares with typical Southern Oscillation changes.

Line 7: "The longitude structure of the Hadley Circulation" I wonder how this relates to the Walker circulation. In principle they could be different things but I suspect that in practice they are related and it is more usual to refer to the Walker circulation.

**Typos**: There are quite a few typos in the manuscript. AI tools are quite good at picking these up. So I haven't listed the ones I spotted.

**Section 2.1**: I wondered how strong the applied heat forcing is. From the description on lines 150-151 my impression is that the average temperature of the SST being forced into the model is 29.0 °C. The average SST in the forced region in the forced run is 28.8°C, so the difference is 0.2°C. The partially ramped forcing in the top 100 metres would force at least this temperature difference in over a depth of 30 + 70/2 = 65 metres. The relaxation time-scale is 2 days = $2 \cdot 10^5$ secs. So the heating rate is $\frac{\rho c_p (0.2)(65)}{2.10^5} = 4.10^6 (65) 10^{-6} = 260 \, \text{Wm}^{-2}$.

Suppose that a 0.5 m/s near-surface current of 100 metres depth advects water of temperature $T+dT$ into one side of an area of length dx and water of temperature $T$ out the other side. The heat flux per unit area is then $\rho c_p dT$ (100)(0.5) / dx. Taking $dT = 2K$ this gives heat flux = $4.10^8$ / dx Wm$^{-2}$. A heat flux of 200 Wm$^{-2}$ would be obtained with dx = $2 \cdot 10^6$ m which is about 20° of longitude. If this is correct interannual variations in zonal advection could make a significant but perhaps not dominant contribution to variations in surface heat flux consistent with those in the experiment. It might be helpful to include a calculation of this sort at some point in the paper.

**Section 3**

Lines 175 – 190: I appreciate this more detailed explanation of the flux shown in figure 1. I think it is OK here but it might appear in the methods section.

Line 216: Insert "overall" between small and reduction?

Lines 270-274: I think this is discussing a region between about 20 and 30 °S. It would be helpful to mention that (I tend to think of the south-eastern Pacific as being further south!) The reduction in surface pressure due to the warmer air follows more directly in my view from hydrostatic balance and the fact that surface pressure gradients are stronger than upper air ones.

**Section 5**

This is an important addition to the paper. My one reservation about it is that it feels like the rest of the paper, particularly the concluding section, has not been adjusted to take into account its results. This is most marked in my view for the pressure changes in the Southern Ocean. My impression is that sections 3 and 4 should emphasise that less because its signal in the ensemble is significantly weaker than it is in the original experiment. This comment can be ignored if the 2 mb change is representative of ENSO / Southern Ocean changes.

**Figure B.1**: I think it is interesting that the El Nino 3.4 index and the forcing region SSTs are very similar during the large El Nino events. This suggests to me that much of the surface water at the equator has been advected southward from the forcing region during the El Nino events. This would provide further support and a (an additional?)

mechanism for the author's hypothesis that the SST in the NECC region influences the development of El Nino events.

M. J. Bell

---

## Author Response (AR1)

**Authors Response**

==========

I would like to thank both reviewers for their comments, which forced me think more about the implications of the study. As my colleagues are all oceanographers, I am a bit disappointed that the strongest results from the work concern convection and the Hadley Cell. In contrast the response of the Cold Pool, a major oceanographic feature which attracts lots of funding, is rather weak. But I have to live with that.

**Reviewer 1**

========

Title: This seems appropriate but it is not very catchy and does not mention El Nino.

[The title has been been changed to that proposed in my original response to the reviewer - document R1]

Abstract.

[The abstract has been revised taking into account the reviewers comments]

Line 7: Longitudinal structure

[No change - following my comments in R1]

Typos:

[I have correct all that the spell checker and I could find]

Section 2.1 Heat Forcing

[Following my comments in R1 I have add some comments emphasising the importance and role of advection in section 2.1.]

Lines 175-190 - Section on Analysis

[As discussed in R1, I have not reorganised the text.]

Line 216.

['overall' inserted in the text.]

Lines 270-274:

[I have added the text "in the region between 230°E to 280°E and between 15°S and 35°S" in section 3.3.1 on Cloudiness]

**Section 5**

[After considering the reviewers comments further I decided that the results of sections 5.2 and 5.3 were really important and needed to be highlighted in the summary. For this reason I have made a major revision to the summary section emphasising the importance of the overlap between the NECC temperature range and the temperatures at which deep atmospheric convection can be triggered in the ITCZ.

The importance of the results from the ERA5 reanalysis are less clear so I have made no further changes.]

Figure B1

[Following the reviewer's comments I investigated the ERA5 data further. However it soon became clear that when major equatorial heating events occur at the height of strong El Ninos, the warmest temperatures on the Equator appear are linked to enhanced temperatures in the South Pacific Convection Zone.

Section B1 now includes a short comment on this.

Following both reviewer's comments I have added a comment on turbulent heat flux from warmer regions near the end of section 4.3]

**Reviewer 2**

Single member Study

[See my published response to Review 2 - document R2]

Length of Paper

[As in R2]

Direct Dynamical Response

[Following both reviewer's comments I have added an extra section on heat flux to the end of section 4.3.]

The abstract and conclusion : El Nino like changes

[There are many possible definitions on El Nino, and I fell into the trap of meaning different things in the abstract and summary.

In the revised abstract I use the phrase "for many of the atmospheric and oceanic features associated with the El Ni\~no and the Southern Oscillation". It is a bit wordy but does not imply that it covers everything.

In the revised summary I also write about "El Nino type changes" for the same reason.]

**Hadley Circulation**

[I appreciate that the I had not emphasised enough the change to the Hadley Circulation and the resulting impacts elsewhere.

I have therefore revised the summary section, i.e. 6.1 and 6.2, to give convection and the impact of the Hadley circulation more emphasis. I also included references to papers (Baines 2006, Li et al 2023) on the longitudinal structure of the Hadley Cell.

For a similar reason, section 6.3 now includes a reference to more local effects (Peng et al 2024).]

Figure 1 : Difference

[A difference plot has been added to Fig. 1]